# Microbial Diversity and Adaptation under Salt-Affected Soils: A Review

Chiranjeev Kumawat [1,*,†], Ajay Kumar [2], Jagdish Parshad [2,*,†], Shyam Sunder Sharma [1], Abhik Patra [3,4], Prerna Dogra [1], Govind Kumar Yadav [1], Sunil Kumar Dadhich [1], Rajhans Verma [1] and Girdhari Lal Kumawat [5]

1   Department of Soil Science and Agricultural Chemistry, Sri Karan Narendra Agriculture University, Jobner, Jaipur 303329, Rajasthan, India; ssharma.soils@sknau.ac.in (S.S.S.); prernadogra.soils@sknau.ac.in (P.D.); yadav.govi004@gmail.com (G.K.Y.); skdadhich.soils@sknau.ac.in (S.K.D.); rajhansverma.soils@sknau.ac.in (R.V.)
2   Department of Microbiology, COBS&H, Chaudhary Charan Singh Haryana Agricultural University, Hisar 125004, Haryana, India; ajaykumar@hau.ac.in
3   Department of Soil Science and Agriculture Chemistry, Institute of Agricultural Sciences, Banaras Hindu University, Varanasi 221005, Uttar Pradesh, India; abhik.patra1@bhu.ac.in
4   Krishi Vigyan Kendra, Narkatiaganj, West Champaran 845455, Bihar, India
5   Department of Plant Pathology, Sri Karan Narendra Agriculture University, Jobner, Jaipur 303329, Rajasthan, India; girdhari.pbg@sknau.ac.in
\*   Correspondence: chiranjeev.soils@sknau.ac.in (C.K.); lect.jagdish@gmail.com (J.P.)
†   These authors contributed equally to this work.

**Abstract:** The salinization of soil is responsible for the reduction in the growth and development of plants. As the global population increases day by day, there is a decrease in the cultivation of farmland due to the salinization of soil, which threatens food security. Salt-affected soils occur all over the world, especially in arid and semi-arid regions. The total area of global salt-affected soil is 1 billion ha, and in India, an area of nearly 6.74 million $ha^{-1}$ is salt-stressed, out of which 2.95 million $ha^{-1}$ are saline soil (including coastal) and 3.78 million $ha^{-1}$ are alkali soil. The rectification and management of salt-stressed soils require specific approaches for sustainable crop production. Remediating salt-affected soil by chemical, physical and biological methods with available resources is recommended for agricultural purposes. Bioremediation is an eco-friendly approach compared to chemical and physical methods. The role of microorganisms has been documented by many workers for the bioremediation of such problematic soils. Halophilic Bacteria, Arbuscular mycorrhizal fungi, Cyanobacteria, plant growth-promoting rhizobacteria and microbial inoculation have been found to be effective for plant growth promotion under salt-stress conditions. The microbial mediated approaches can be adopted for the mitigation of salt-affected soil and help increase crop productivity. A microbial product consisting of beneficial halophiles maintains and enhances the soil health and the yield of the crop in salt-affected soil. This review will focus on the remediation of salt-affected soil by using microorganisms and their mechanisms in the soil and interaction with the plants.

**Keywords:** halophilic bacteria; PGPR; arbuscular mycorrhizal fungi; cyanobacteria

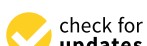

## 1. Introduction

The enhancement in crop productivity in proportion to the growing population for feeding has been a big challenge since the inception of agriculture. The Global Agricultural Productivity (GAP) (2018) index states that the fulfillment of the food demands of a population of 10 billion in 2050 is not possible at the current growth rate of food production [1]. Soil supports the sustainable survival and development of humans, along with air and water. Food security, water scarcity and environmental pollution are the most serious challenges for all people. Crop productivity is affected by many abiotic factors which include temperature, soil pH, pesticides and fertilizer application, heavy metal, drought and salinity [2] The global scarcity of water resources, environmental pollution and the

increased salinization of soil and water are important issues at the beginning of the 21st century. Soil also faces different stresses such as heat stress, drought stress and salt stress. These soil stresses are responsible for a significant reduction in crop yield. Salt stress is one of the important stresses which plays important role in plant growth and development. Salt-affected soils occur all over the world, especially in arid and semi-arid regions. Globally, there are 1 billion ha of salt-affected soil, and in India, nearly 6.74 million $ha^{-1}$ of the area is under salt-affected soil [3]. The increasing rate and expansion of areas under salinity stress have created an insecurity of food demands in many countries. The salinization of coastal belts in the delta regions of India, Myanmar and Bangladesh, which majorly contribute to world rice production, are facing danger to food security [4,5]. Irrigated salinity-stressed areas have caused USD 12 billion in global income loss annually [6]. Large areas in the Indian states of Rajasthan and Gujarat comprise saline-unproductive land in the form of saline lakes, salt depressions and saline swampy lands devoid of any vegetation or supporting very meager cover. The salt-stressed condition negatively affects important soil activities such as nitrification, respiration, microbial diversity, mineralization residues decomposition, etc. [7]. High fertilizer application also results in soil salinity and deteriorates crop productivity due to the imposition of an osmotic regulation, causing water extraction for plant growth and development [8,9]. Soils having excess salt on the soil surface and in the plant root zone in such an amount can retard the growth and development of plants. These soils are distributed relatively more extensively in the arid and semi-arid regions as compared to the humid regions [9]. The reclamation and management of such soils require specific approaches for long-term productivity. Physical and chemical processes have long been performed in the reclamation of saline soil. Physical process such as flushing, leaching and scraping, along with neutralizing agents such as gypsum and lime under alkali and acid soil, are practiced under chemical methods for the removal of soluble salts [10]. Salt-tolerant crops such as barley and canola are grown, however, due to a normal salt-tolerant ability; these crops could not reach the world level and were not able to perform under high salt concentrations [11]. Morton et al. (2019) has reported that despite vigorous efforts from researchers, only a few salt tolerance genes have been identified as having real applications in improving the productivity of saline soils [12]. A major focus in the coming decades would be on safe and eco-friendly methods by exploiting the beneficial micro-organisms in sustainable crop production [13]. The inoculation of some naturally occurring microbes in the soil ecosystem advances soil physico-chemical properties, soil microbial biodiversity, soil health, plant growth and crop productivity [14]. In the recent past, researchers have demonstrated that the use of halophilic plant growth-promoting rhizobacteria enhanced crop productivity and soil health [15]. So, this review will focus on the different types of microorganisms such as bacteria, fungi, mycorrhiza, cyanobacteria, etc., which are capable of the bioremediation of salt-affected soil.

## 2. Ecology of Saline Soil Microorganisms

The communities of microbial diversity play an important role in the nutrients cycling. Environmental stress in the soil affects the microorganism and becomes detrimental to the survival of the microbes, decreasing the activities of surviving cells because of the metabolic load imposed by the starting and activation of the stress-tolerant mechanisms [16–19]. Under a dry and hot environment where low humidity and soil salinity are the most stressful factors for soil microbial diversity, the activity and metabolism of the microorganisms decrease. The detrimental and negative effects are more in the rhizosphere of the plant because of the increase in the water absorption by the plants due to transpiration. Life under the stress of salinity has a requirement of high bioenergetics because the microflora need to maintain the osmotic equilibrium between the cytoplasm of the microbes and the surrounding environment. Microbes under salt stress conditions survive by excluding the sodium ions from the cell inside, so microorganisms require a high energy, which is sufficient for osmoregulation [20,21]. Cells are separated by the medium using a cell membrane, which is permeable to water. When the concentration of the salt increases

in the surrounding medium of the cells and reaches a point where the solute concentration becomes high, the solute concentration inside the cells loses the water and leads to the risk of the drying out of the cell. Cells can tolerate the salt counterbalances that increase in the osmotic pressure. Microbes had to be able to survive at high salt or solute concentrations in the medium in order to maintain an equally high concentration of solute in the cell cytoplasm. The rising of the solute concentration in the cell cytoplasm can be achieved by the synthesis and accumulation of the small organic molecules, which are called compatible solutes because of their non-interference with cellular functions [22].

The accumulation of the potassium ions ($K^+$) inside the cell cytoplasm is another short-term response strategy to escape in situations where the salt concentration has rapidly increased. The enzymatic process is affected by the high ions concentration; this is why most organisms synthesize the small organic molecules. Compatible solutes are accumulated in the cells, whereas the salt ions are toxic, as they interfere in the enzymatic activities, and sodium ($Na^+$) and chlorine ($Cl^-$) must be excluded from the cells. The exclusion of the salt ions is possible through the cross-membrane protein pump. The binding of the $K^+$ ions is responsible for the activation of more than 50 plant enzymes, so an increase in the concentration of salt or $Na^+$ interferes with the binding of the $K^+$ binding sites, which leads to the disruption of the metabolic processes [23].

The high concentration of sodium ($Na^+$) ions results in the retardation of plant growth and produces necrosis symptoms in plants. A high concentration of $Cl^-$ leads to a lack of chlorophyll by degrading it [24]. The high concentration of salt restricts the limit for the uptake of the water by the plant roots against the negative soil water potential. High salt concentrations also result in an imbalance in the uptake of the plant nutrients in the rhizosphere. The exposure of the microorganisms to the salt stress conditions changes the expression pattern of the RuBisCO enzyme, which helps in carbon dioxide ($CO_2$) fixation and makes carbon compounds for energy synthesis and other reactions available. The different types of osmo-tolerant proteins are produced during harsh conditions, which helps in the water holding and helps plants to tolerate the exposure to salt stress levels.

The salt tolerant microbes are divided in to four groups by Kushner (1993), i.e., non-halophilic <0.2 M NaCl, slight halophilic 0.2–0.5 M NaCl, moderate halophilic 0.5–2.5 M NaCl and extreme halophiles >2.5 M NaCl. The halo-tolerant microorganisms can tolerate high salt concentration but grow best in media containing <0.2 M (1%) salt. This definition of "halo-tolerant" is widely accepted [25–28]. The saline soil consists of an abundance of halophilic microorganisms in the soil and most dominantly belong to the genera of *Bacillus*, *Pseudomonas*, *Micrococcus* and *Alcaligenes* [29]. Garabito et al. (1998) investigated the saline soil situated in different locations of Spain, where he isolated 71 microorganisms for halo-tolerant, gram-positive, endospore-forming and rod-shaped *Bacillus* genus [30]. The salinity affects the composition of the microbial diversity [19,31–34]; thus, the microbial genotypes are different in their tolerance to a low osmotic potential [34,35]. A low osmotic potential results in a decrease in the spore germination and growth of the hyphae and a variability in the morphology [36] and gene expression [37]. Fungi seem more sensitive towards the salinity environment and osmotic stress than other microorganisms [31,38,39]. Salinity in soil with different concentrations of NaCl resulted in a significant reduction in the total fungal count. Similarly, if the salinity level is >5%, then the number of bacteria and actinobacteria is drastically reduced [40].

The accumulation of the ions that are necessary for the metabolism of cells occurs in halo-tolerant microbes. The other mechanism of cell adaptation in salt stress conditions is the production of organic compounds that will neutralize the concentration gradient between the cell cytoplasm and soil solution. This mechanism of the adaptation results in the higher physiological activities of the microbial community and the consequences. The cell reduces the utilization of the substrate. A better understanding of the changes in the microbial biomass and its activity under salt stress conditions can be achieved by the consideration of water potential (osmotic potential, matrix potential), particularly low water content where the salt concentration increases in the saline soil. Electrical conductivity is

an indicator of microbial stress under salt stress conditions. The microbial biomass is an important labile fraction of the soil organic matter (OM), which acts multifunctionally as an agent of the recycling and transformation of the soil nutrients and OM and also acts as a source of plant nutrients. Microbial secretion is also an important source of the enzyme which helps in the regulation of many mechanisms in soil. The nutrients available for the plants are regulated by rhizospheric microbial activity [19]. So many factors in the soil which affect the microbial community and its function influence the availability of the nutrients and the growth of plants.

The recycling of nitrogen (N) such as mineralization and the immobilization through microbial responses plays an important role in plant growth and development [41]. Nitrogen mineralization is the conversion of the organic nitrogen to an inorganic form of nitrogen, and immobilization is the reverse of mineralization. Both mineralization and nitrification were significantly retarded in the presence of NaCl; maximum inhibition occurred with 4000 mg NaCl kg$^{-1}$ of soil. The inhibitory effect of NaCl on N mineralization was relatively higher in soils treated with $NH_4^+$. The results of this study suggest a greater sensitivity to NaCl by microorganisms that have assimilated $NO_3^-$ [42]. Moreover, the presence of the NaCl retards the immobilization of the N.

## 3. Interaction of Plants and Microbes in Salt-Affected Soils

When the microorganisms are exposed to the high-osmotic environment, rapid fluxes of the cell water out of the cell take place, resulting in a reduction in the turgor and the dehydration of the cytoplasm. Different types of adaptation have been achieved against the outflow of the cell water. The osmotic equilibrium between the cytoplasm of the cell and the surrounding media is maintained by exposing the cytoplasm to high ionic strength. The first response of the cell to the osmotic upshift results in the efflux of cellular water, the uptake of $K^+$ and the accumulation of the compatible solutes into the cell [43].

Salt stress (50–200 mM NaCl) in the legume crops restricts productivity because of the negative and adverse effects on the growth of the root nodule bacteria and host plant, the symbiotic development and the nitrogen fixation ability [44]. A decrease in nitrogen fixation by affecting nodule development and the symbiotic association in *Vicia faba* was observed under the salinity stress in cultural media [45]. However, after the full development of the root nodule under stress-free conditions, the nitrogen fixation continues even after the treatment of salt-stress conditions. The early prolific variety of *Phaseolus vulgar* is tolerated at low levels (48 mM NaCl) but not at higher levels (72 and 96 mM NaCl) of salt [46]. The strain GRA19 of *Rhizobium leguminosarum* biovar. *Viciae* was found to be tolerant to low levels of salt (50 mM) by comparing the growth under stress conditions to that in the absence of stress. Moreover, the growth of symbiotic $N_2$ fixation (acetylene reduction activity) under saline conditions of the faba bean cultivar Alameda inoculated with GRA19 was reduced [47]. The same species of *Rhizobium* vary in terms of their salt tolerance, the tolerance of different species of *Rhizobium* to NaCl ranging from 100 to 650 mM [48,49].

Rhizobia show a marked variation in salt tolerance. A number of strains are inhibited by 100 Mm of NaCL salt [50–52], but growth at salt concentrations of more than 300 mmol has been reported for the strains of *Sinorhizobium meliloti* [53,54] and *Rhizobium tropici* [55]. Some alfalfa, *acacia*, *prosopis* and *leucaena* strains will tolerate 500 mmol$^{-1}$ NaCl [52,54].

## 4. Application Strategy of Halophilic Microbes

### 4.1. Halophilic Bacteria

These halophilic bacteria are capable of balancing the osmotic pressure in the environment. Moreover, the organisms that can survive in highly saline conditions and require salt for proper growth and development are called halophiles. They are very diverse, belonging to three domains of life, i.e., Bacteria, Eukarya and Archaea. They are inhabitants of the soda lakes, salt ponds and rock salt crystals as dormant cells [56,57]. There are two sorts of organisms: those that can tolerate salt and those that require salt for growth and

development [58]. Halotolerance is a mechanism through which halophilic bacteria can maintain growth and development under salinity conditions.

The halophiles are classified as slight halophiles, moderate halophiles and extreme halophiles. Slight halophiles can grow optimally between a 0.2–0.0.5 M (1–3%) NaCl concentration. Moderate halophiles can grow with a 0.5–2.5 M (3–15%) NaCl concentration, and extreme halophiles are able to grow with a 2.5–5.2 M (15–30%) NaCl concentration. Halophiles are aerobic, anaerobic, heterotrophic, phototrophic and chemoautotrophic types found in different environments [59].

In agriculture, plants face various environmental abiotic stresses such as droughts, chilling salinity, nutrient deficiency, pathogens, heavy metals, etc. This stress problem leads to abnormalities in the growth and development of the plants. Due to low rainfall, high temperatures and poor-quality water in arid and semiarid areas, soil faces the salinity problem, which is considered as a major environmental stress [60]. Halophilic bacteria adapt to salinity by a different method, assisting the plant in surviving under salt stress circumstances. Plants have various biochemical and physiological strategies to live in salt-stressed soil, such as osmolyte production, antioxidant enzymes, hormones and ion exclusion. Aside from all of these plant defense systems, the bacterial community in the soil, such as halophilic bacteria, also plays a significant role in increasing salt tolerance in the soil.

### 4.2. Taxonomy of Halophilic Bacteria

Halophilic microorganisms are salt-loving organisms that belong to the order *Halobacteriales* and to the family *Halobacteriaceae*. The first halophilic microorganism was discovered in Utah's Great Salt Lake and was called *Halanaerobium praevalens*, which was described and classified as a genus in the *Bacteroidaceae* family [61].

After that, new halophilic bacterial species and genera were identified based on 16S rRNA sequencing and the lipid profiling of the membrane. Different halophilic species have been listed in Table 1.

**Table 1.** Halophilic bacteria species with the salt-tolerant range.

| Halophilic Bacterial Species | Salinity Range for the Growth and Development (%) | References |
|---|---|---|
| *Kangiella spongicola* | 2–15 | [62] |
| *Halanaerocella petrolearia* | 6–26 | [63] |
| *Salisediminibacterium cookie* | 3–30 | [64] |
| *Amphibacillus cookie* | 6–26 | [65] |
| *Desulfohalophilus alkaliarsenatis* | 12.5–33 | [66] |
| *Halanaerobacter jeridensis* | 6–30 | [67] |
| *Natribacillus halophilus* | 7–23 | [68] |
| *Fodinibius salinus* | 10–15 | [69] |
| *Alkalibacterium gilvum* | 0–17.5 | [70] |
| *Halomicroarcula pellucida* | 20–30 | [71] |
| *Salinibacter iranicus* | 12–30 | [72] |
| *Halanaerobium sehlinen* | 5–30 | [73] |
| *Saliterribacillus perciscus* | 0.5–22.5 | [74] |
| *Limimonas halopajila* | 15–30 | [75] |
| *Aquibacillus halophilus* | 0.5–20 | [76] |

**Table 1.** *Cont.*

| Halophilic Bacterial Species | Salinity Range for the Growth and Development (%) | References |
|---|---|---|
| *Halobellus salinus* | 15–30 | [77] |
| *Bacillus daqingensis* | 0–16 | [78] |
| *Oceanicola flagellatus* | 0–21 | [79] |
| *Spiribacter salinus* | 10–25 | [80] |
| *Halomonas huangheensis* | 1–20 | [81] |
| *Salifodinibacter halophilus* | 25 | [82] |
| *Halomonas sambharensis* | 5–8 | [83] |
| *Lentibacillus saliphilus* sp. nov. (type strain YIM 93176[T]) | 0–22 | [84] |
| *Halomonas urmiana* sp. | 0.5–20 | [85] |
| *Marinobacter halodurans* sp. nov. | 1–18 | [86] |
| *Aliifodinibius saliphilus* sp. nov. | 3–25 | [87] |
| *Arhodomonas recens* | 2–25 | [88] |

*4.3. Adaptability Mechanisms of Halophilic Bacteria for Saline Environments*

Water is the prime element which is the responsible for life. Living microorganisms have the adaptation ability to survive under adverse environments. Microorganisms that have not adapted to saline conditions will lose water, causing the cells to shrink and eventually die due to a lack of cellular structure and function. To avoid excessive water loss in such conditions and preserve cellular structure and function, halophilic bacteria have evolved two sorts of techniques to deal with high salt concentrations [89]. The first strategy is the salt-in strategy, while the second is the compatible solute strategy. Bacterial cells keep the internal and exterior environments osmotically equal by collecting a high concentration of KCl. This method is carried out by the cell by changes in various physiological metabolisms such as enzyme activity, cellular component production and the shape and function of some organelles. The high-salt-in method protects halophiles from a saline environment by accumulating inorganic ions intracellularly to keep the salt concentrations in their environment balanced. Bacterial cells keep the internal and exterior environments osmotically equal by collecting a high concentration of KCl. Halophiles consist of the $Cl^-$ pumps and transfer $Cl^-$ from the environment into the cytoplasm in this process. To enhance the uptake and release of $Cl^-$, arginines and lysines are placed at both ends of the channel [90].

Most of the halophilic microorganisms protect the cell from high salt concentrations by the accumulation of compatible solutes such as organic (proline, betaine, ectoine, trehalose) and inorganic solutes ($K^+$, $Mg^{2+}$, $Na^+$) [91,92]. The osmolytes or compatible solutes are released in the cytoplasm by the bacterial cell itself or they are taken from the medium. Most of the bacteria lack the intracellular system for the active transport of water to nullify the external osmotic pressure. Therefore, the internal environment is maintained by the transport/synthesis of a group of compatible solutes without affecting the metabolic function of the cell [93,94]. According to the chemical nature, compatible solutes are classified as anionic solutes, zwitterionic solutes and non-charged solutes. Organic anions are used to balance the internal environment of the halophilic bacteria under high salty conditions. Halophilic bacteria such as Halomonas and Halobacterium synthesize ectoine and L-glutamte, respectively, to survive under the salinity-stressed conditions [95]. Some halotolerant bacteria including Bacillus, Pseudomonas, Aeromonas and Zymomonas use the polyols compounds such as sorbitol, arabitol, glycerol and mannitol for osmoadaptation under salt-stressed conditions [96]. Halophilic bacteria use neutral amino acid-derived zwitter solutes as osmolytes in salt-stressed conditions [97]. Betaine is a natural compound

with a negative charge used as an osmolyte for the protection of cells in order to cope with high osmotic stress by maintaining an internal balance by the regulation of water inside the cell. Different halophilic bacteria such as Halomonas, Virgibacillus, Oceanobacillus and Polaribactercan synthesize betaine from the glycine with the primary amine methylated to form a quaternary amine. Some methanogens such as methanohalophilus and methanohalobium can accumulate and synthesize betaine by the methylation of glycine or choline oxidation [98–100]. Ectoine (cyclic tetrahydropyrimidine), which is either accumulated from the external environment or synthesized from the medium, is used as the osmolyte by the halophilic bacteria to protect against the salt-stressed conditions. This was detected from the Halorhodospora halochloris bacteria, which was isolated from the hypersaline Mono lake [101]. Ectoine osmolytes have been found in halotolerant and halophilic bacteria such as Halomonas, Oceanobacillus, Nesterenenkonia, Methylophaga and Methyllarcula [94,102,103]. Some polar and non-charged organic molecules have also been used as osmolytes to protect the cell from high salt-stressed conditions. Glycerol osmolyte has been detected in some bacteria and halotolerant yeast under salt-stressed conditions [93,104]. Some sugar molecules such as trehalose have been detected in the halotolerant and halophiles and have been used as compatible solutes to cope with dessication, heat, cold and a hypersaline environment. Some proteobacteria and marine cynobacteria are known to accumulate sucrose as an osmolyte in salt-stressed conditions. Some proteins such as proline, acetylated glutamine dipeptide and carboxamine also act as the osmolytes and protect the cell from high salt conditions. They are mostly found in halophilic purple sulfur bacteria and marine phototrophic bacteria [89] (Figure 1).

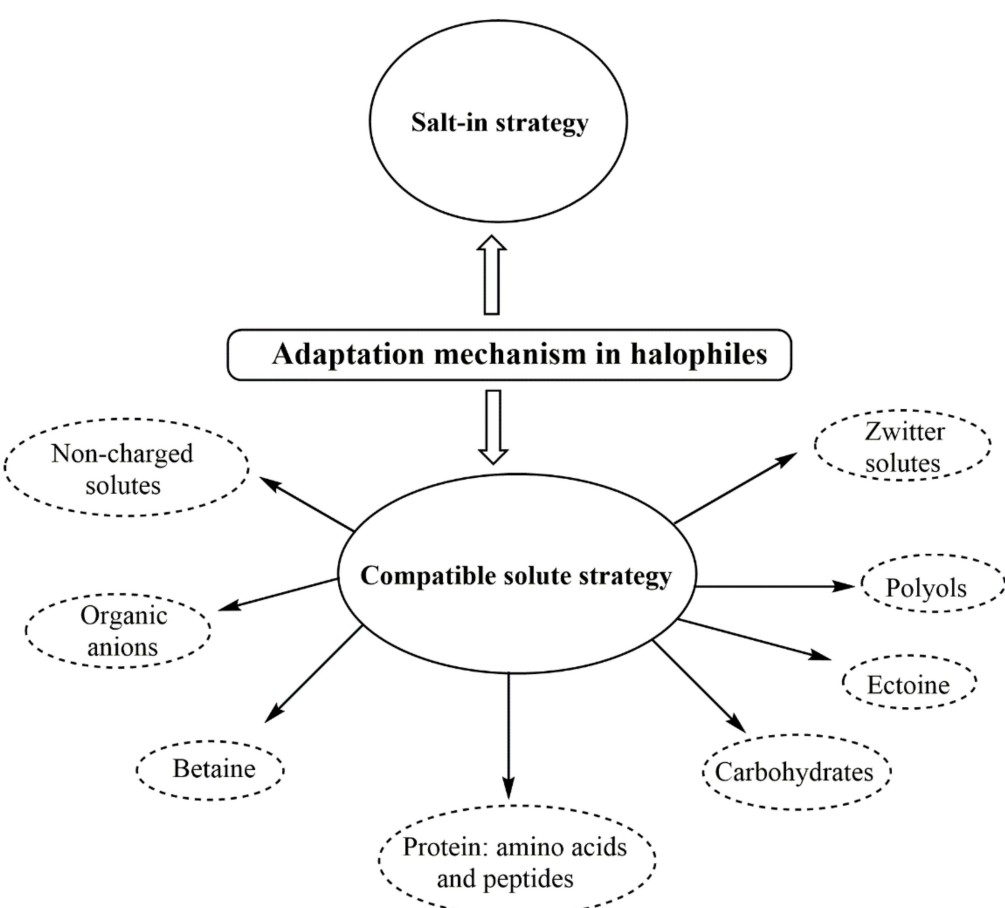

**Figure 1.** Different mechanisms of adaptation in saline conditions by halophilic bacteria.



## 5. Halophilic Bacteria: Role of Halophilic Bacteria in Plant Growth Promotion under Salt Stress

During growth and development, each living organism or plant is subjected to the harsh conditions of the soil. To escape the stress circumstance, they will either fight or devise an alternative approach. Because plants are highly delicate and sessile, they cannot escape the bad conditions; thus, they fight back against them. With the aid of multiple mechanisms, halophilic bacteria boost their tolerance capacity, development and production and overcome the detrimental impacts of abiotic stress conditions with specific functional features.

### 5.1. The Role of Bacterial Phytohormones

Bacterial phytohormones are organic compounds that have a low concentration and impact the physiological and biological processes in plants. These tiny quantities of bacterial phytohormones influence the control of several processes involved in plant differentiation and development. Bacterial hormones, which are plant growth hormones secreted near the plant roots, can initiate a physiological response in the host plant. Plant growth-promoting bacteria (PGPB) generate phytohormones such as IAA, cytokinins, abscisic acid, gibberellins and other growth regulators that aid in plant growth and development. All of these phytohormones prolong root stimulation by dramatically increasing root length and surface area, which leads to increased nutrient absorption and hence enhances plant health in salt-challenged circumstances [105].

### 5.2. Aminocyclopropane-1-Carboxylate (ACC) Deaminase

In extremely low quantities, ethylene is an essential and volatile bacterial phytohormone that impacts plant growth regulation. Ethylene phytohormones influence the growth of plant vegetative parts, the rooting of cuttings and nodulation [106], as well as the transmission of signals for the response to salt stress surrounding the root zone [107]. The overproduction of ethylene hormones in response to abiotic stress situations can limit plant growth and development. Chemical inhibitors such as cobalt ions and aminoethoxyvinylglycine are commonly used to overcome these difficulties. However, these compounds are prohibitively costly and hazardous to the environment. Salt-tolerant bacteria can generate aminocyclopropane-1-carboxylate (ACC) deaminase, which converts ACC to $\alpha$-ketobutyrate and ammonia, lowering the ethylene levels in salt-stressed plants [108].

### 5.3. Phosphate Solubilization

Phosphorous (P) is an important macronutrient that is required for the production of many biochemicals such as nucleotides, phospholipids, nucleic acid and phosphoprotein, as well as for plant growth and development. Under salt stress circumstances, the availability of phosphorus decreases, and signs of P shortage develop [109]. Organic and inorganic phosphorus are the two types of phosphorus present in soil. The mobility and availability of P to plants are quite low in comparison to other nutritional elements such as zinc, iron, copper, potassium and so on [110]. The majority of the phosphorous in the soil is in the insoluble form, making the mobility and availability of the P to the plant difficult or impossible. Halophilic strains aid in the conversion of insoluble P to soluble P and in the maintenance of soil P levels. A lot of research has been done on halotolerant strains that can solubilize and make phosphorus available. The phosphorus mobilization and absorption were demonstrated in the blackpaper, which resulted in increased root proliferation and plant growth [111]. Rhizobacterial strains can thrive in high salt conditions (60 g/LNaCl) and are effective P solubilizers in soil [112]. Under salt stress conditions, the *pseudomonas* strains had a substantial influence on the growth and development of *Zea mays* L. [113]. Under saline circumstances, PSB *Herbaspirillum seropedicae* and *Burkholderia* sp. inoculation increased crop weight by 1.5–21 percent [114].

### 5.4. Antioxidative Activity

The salt stress state induces the creation of reactive oxygen species, which destroys various biomolecules such as proteins and lipids and causes plant death [115]. Plants contain antioxidant processes that allow them to live in the presence of ROS [116]. There are many antioxidative enzymes (superoxide dismutase, peroxidase, and catalase) and non-enzymatic antioxidants (ascorbic acid, glutathione) that aid in the ROS scavenging processes [117]. Several halotolerant PGPR, such as *S. proteamaculans* and *Rhizobium leguminosarum*, are known to produce these enzymes (SOD, POX, CAT) and aid in the plant's survival under salt stress conditions. It was recently discovered that salt-tolerant bacteria (*P. simiae* AU) enhance peroxidase and CAT gene expression in soybean plants following 100 mM NaCl of stress inoculation [118]. PGPR inoculation mitigates the harmful effects caused by the oxidative stress by enzymatic and non-enzymatic mechanisms under saline-stressed conditions. In the case of non-enzymatic mechanisms, they reduced the exposure to ROS by migrating to less solar radiation space. The pigment production and packaging of the DNA with proteins and chromatins provide alternate sites for the attack of reactive oxygen species. Some non-enzymatic antioxidant compounds also prevent reactive oxygen species. On the other side, the enzymatic method produces different enzymes such as superoxide dismutase, catalases, glutathione peroxidases, peroxiredoxins, etc. without generating more reactive species. Antioxidant enzymes transform the harmful products into less harmful molecules or locate them and degrade them. These methods also maintain the appropriate physiological levels of the reactive species such as ROS [119].

### 5.5. Siderophore Producers

Iron is an essential nutrient for plant growth and development because it functions as a cofactor in several metabolic processes and redox activities. Because the insoluble form of iron (ferric hydroxide) is present in the soil, it is not accessible to the plant and acts as a limiting nutrient for the plant's growth and development. Abiotic stress causes iron to be unavailable, making microbe acquisition a significant issue [120]. Several bacteria with particular mechanisms are present, which help to solubilize the iron nutrient and make it accessible to the plants. Halotolerant bacteria assist the plant in surviving under salt stress conditions and increase iron availability through the synthesis of siderophores. Siderophores are tiny molecular weight compounds that chelate iron and transfer it into cells [121–123]. *Halobacillus trueperi* MXM-16 and the *Chromocurvus halotolerans* strain EG19 produced the siderophores that were hydroxamate in nature [124,125].

## 6. Arbuscular Mycorrhizal Fungi

Arbuscular mycorrhizal fungi are fungi that have a symbiotic relationship with terrestrial plant roots (AMF). Many scientists have studied the effect of mycorrhiza in plants' adaptation to salt stress conditions. Mycorrhizal inoculation influences the ionic balance, nutrient solubilization and mobilization, photosynthesis efficiency, physiological and biochemical performance on plant development and helps to decrease salt tolerance (Figure 2) [126].

Arbuscular mycorrhizal fungi improved salt stress tolerance in the host plant by (a) increasing nutrient and water mobilization and uptake by the extensive hyphal network [127–129], (b) changing the plant morphology and physiology, allowing the plant to adapt to salt stress [130], (c) plant hormone production and (d) the interaction of mycor [131–134].

This AMF connection with plants improves water and nutrient intake, solubilizes nutrients and aids in nutrient cycling in soil, root architecture and the provision of vital nutrients to host plants under salt stress. Mycorrhizal fungi play a crucial role in ion regulation and membrane transport proteins, which govern the host plant's ion homeostasis. As a result, it is clear that AMF association with plants considerably enhances the concentration of macro and micronutrients [135].

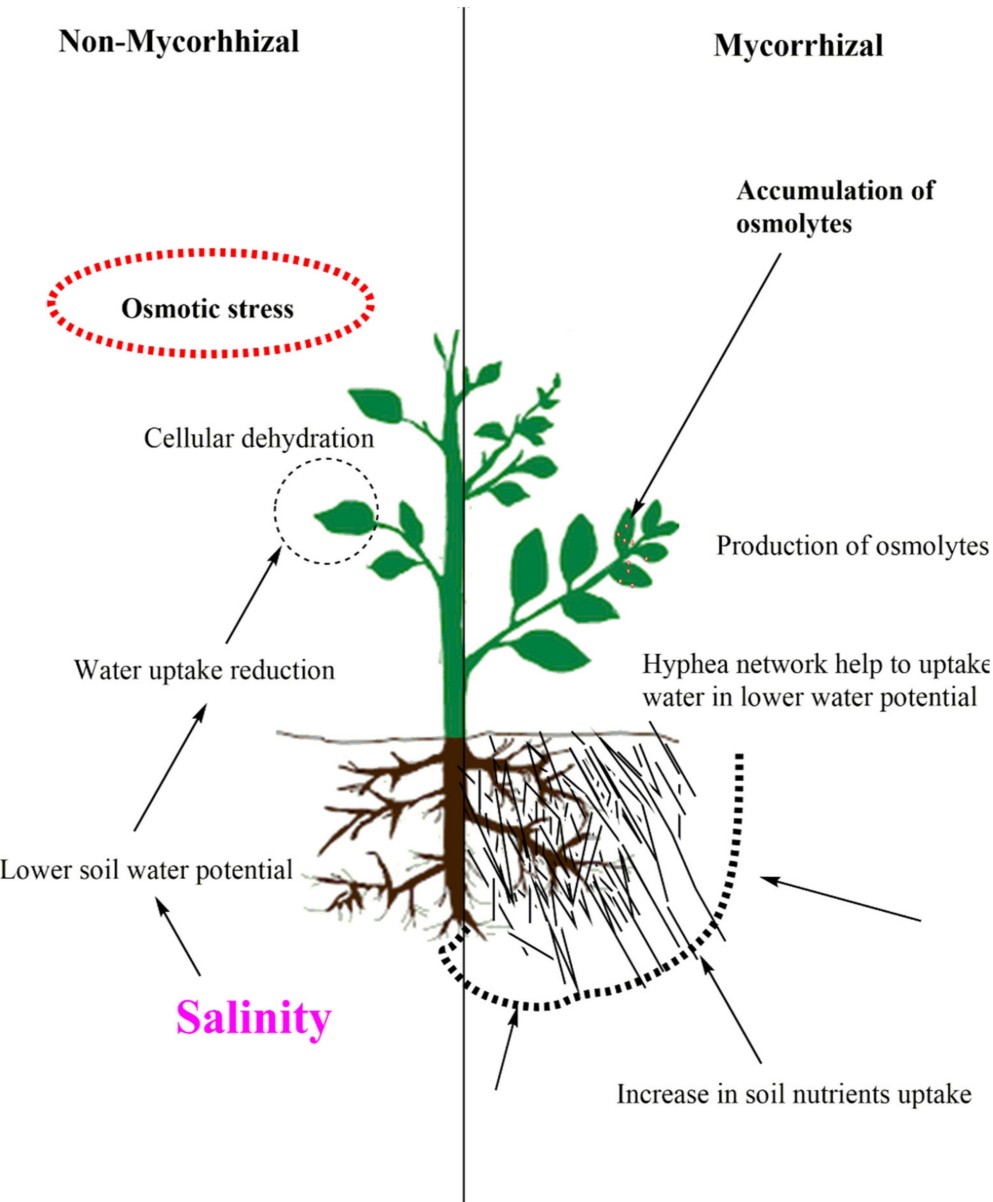

**Figure 2.** Salt stress tolerance by the mycorrhizal fungi compared to the non-mycorrhizal fungi.

It has been discovered that AMF colonization considerably increases the chlorophyll content of numerous plants, including *Solanum lycopersicum* L. and lettuce [136,137]. Plants exposed to salt stress evolve several unique defensive mechanisms, such as increased osmolyte synthesis and antioxidant enzymes, to protect themselves from oxidative damage [138–140]. The AMF relationship dramatically increased antioxidant activities such as peroxidase, catalase, superoxide dismutase and others. During the early phases of the salt treatment, mycorrhization boosted the activity of numerous antioxidants, including superoxide dismutase (SOD), peroxidase (POD), ascorbate POD (ASA-POD) and catalase (Cat). Hajiboland et al. (2010) and Huang et al. (2010) investigated how AMF interaction with plants mitigates the oxidative stress caused by salt stress conditions by boosting antioxidant synthesis and scavenging reactive oxygen species (ROS) [136,141].

The proline content of salt-stressed AMF infected peanuts was similarly increased [142]. Sannazzaro et al. (2007) discovered proline and polyamine accumulation in two genotypes of Lotus glaber after inoculation with *Glomus intraradices*. Under salt-stressed circumstances, proline production was also seen in mycorrhiza-inoculated *Cyamosis tetragonoloba* and *Glycine max* [143,144].

Arbuscular mycorrhizal fungi, in addition to the factors mentioned above, play an important function in improving soil quality and health. Glomalin, a glycoprotein produced by AMF hyphae in the soil, aids in soil aggregation [145]. Although the precise process or gene responsible for glomalin synthesis is unknown, various studies have shown that glomalin and its related soil proteins generated by AMF might contribute to the construction of a "sticky" string bag of hyphae that would stabilize aggregation [146,147].

The favorable effect of mycorrhizal fungus on maize and cotton development under salt stress conditions was related to an increase in proline synthesis and phosphorus absorption [148]. Table 2 shows the responses of plants to AMF inoculation on host species subjected to salt stress treatments.

**Table 2.** Response of the plants to AMF inoculation under salt stress conditions.

| Host Plant | Fungal Species (AMF) | Response by Plant | References |
|---|---|---|---|
| *Cucumis sativus* L. | *Glomus etunicatum, Glomum, intraradices, Glomus mosseae* | Biomass increased, photosynthesis pigments synthesis, antioxidants enzymes increased | [149] |
| *Solanum lycopersicum* L. | *Rhizophagus irregularis* | Enhanced leaf area, leaf number, root and shoot dry weight and growth harmones | [150] |
| *Oryza sativa* L. | *Claroideoglomus etunicatum* | Quantum yield of PSII and photosynthetic rate increased | [151] |
| *Aeluropus littoralis* | *Claroideoglomus etunicatum* | Enhanced root, shoot dry mass, soluble sugars, free amino acids | [152] |
| *Solanum lycopersicum* L. | *Glomus intraradices* | Improved dry matter, growth parameters, chlorophyll content and ions uptake | [136] |
| *Acacia nilotica* | *Glomus fasciculate* | Enhanced root, shoot dry mass, P, Zn and Cu content | [153] |
| *Leymus chinensis* | *Glomus mosseae* | Increase in the colonization rate, seedling weight, water content, P and N | [154] |

## 7. Cyanobacteria

Cyanobacteria are prokaryotic microorganisms that are capable of carbon (C) and nitrogen (N) fixation. Cyanobacteria or blue-green algae (BGA) provide 25–30% N ha$^{-1}$ season$^{-1}$ in rice fields [155].

Blue-green algae also improve soil health by providing extracellular carbohydrates, secondary metabolites and hormones. Cyanobacteria increase the soil porosity and water holding capacity of degraded soil due to the soil salinity and high chemical fertilizers application [156]. Eight BGA species such as *Nostoc*, *Anabaena*, *Calothrix* and *Aulosira* have been selected for field evaluation against pH, salinity and dessication in coastal areas of Orissa. Blue-green algae also help in the amelioration of sodic soil, as cyanobacteria are able to tolerate high sodium concentrations in wet seasons.

Halo-tolerant cyanobacteria (*Nostoc calcicola*) and the possible salt-tolerant mechanism are depicted in Figure 3 [157]. A study stated that the inoculation of cyanobacteria and gypsum changes the soil properties, which indicates the reclamation of the salt-affected soil. Cyanobacterial-treated soil showed significant decreases in pH, EC and Na$^+$, and the organic carbon (OC) content increases significantly. A combination of *Nostoc calcicola* and gypsum seems effective for the treatment of the saline-alkaline soils. Some cyanobacteria release the cyanotoxin under stressed conditions and may have an impact on the seed germination and plant growth, but the phytotoxicity is concentration-dependent, and the field study of the phytotoxicity is inadequate [158].

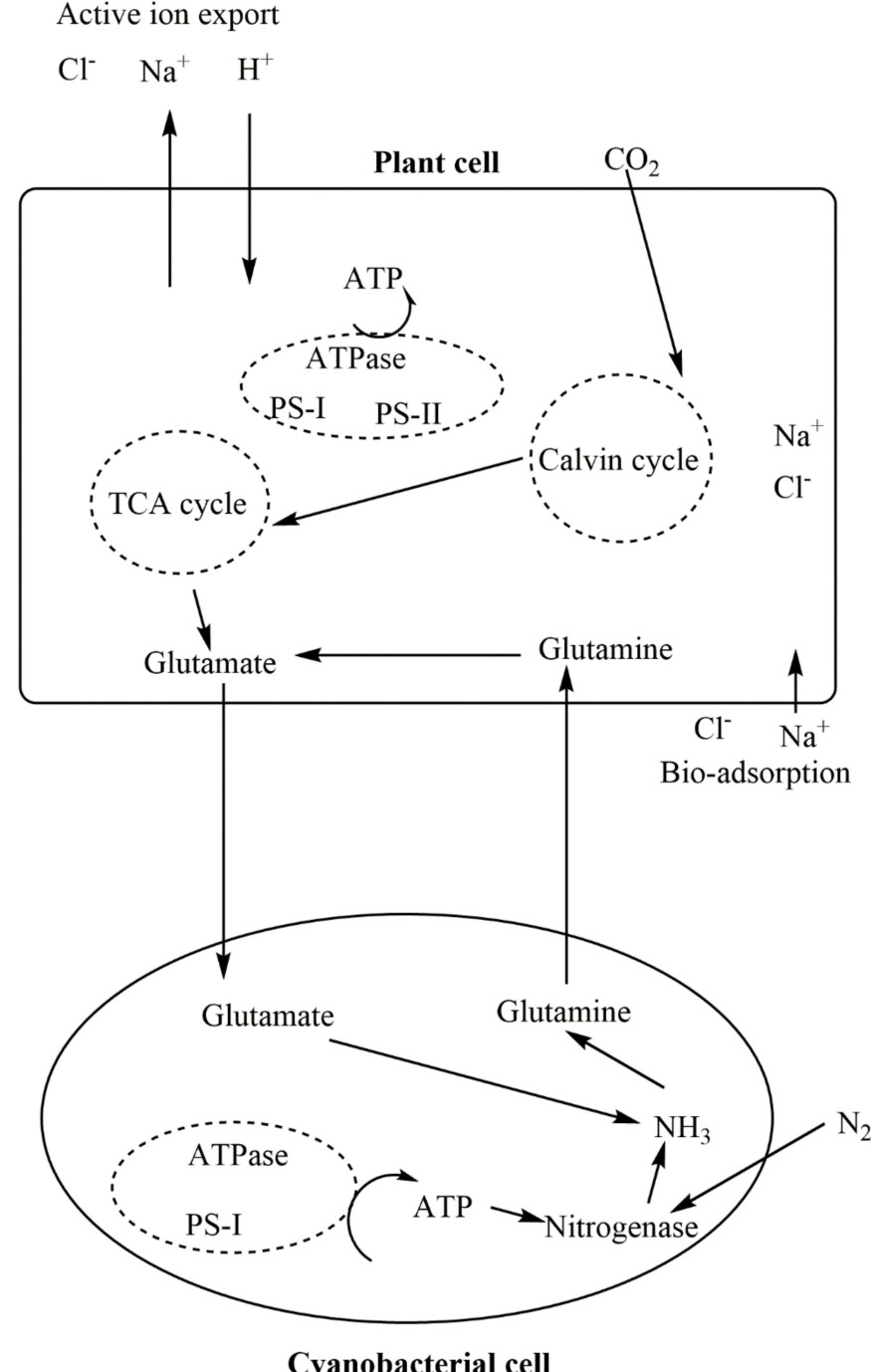

**Figure 3.** Possible mechanisms of salt stress tolerance and salt-affected soils remediation by cyanobacteria.

Alkaline soils, which have high $Na^+$ contents and pH values, enhance the growth of N-fixing cyanobacteria, with a significant decrease in pH. Different types of the organic metabolites released by cyanobacterial activities in the soils also help in maintaining the soil fertility year after year [159]. The addition of *Nostoc calcicole* to the saline/alkaline stress soil reduces the pH content, indicating the improved soil fertility. The dominant growth of *Nostoc calcicola* in saline/alkaline-stressed soils might be because of the salt tolerance capability, which suggests that *Nostoc calcicole* could be a good biological approach for soil reclamation. Singh (1961) recommended that BGA application can be effective for the reclamation of alkaline soils, as they are able to grow on these conditions, while other plants suffer to grow on them [160]. Pandey et al. (2005); Jaiswal et al. (2010) and Murtaza et al.

(2011) have also suggested the role of cyanobacteria in the reclamation of saline-alkaline soils [161–163].

## 8. Plant Growth-Promoting Bacteria

Salt-affected soil is becoming more of a concern over time, and it might be the result of a natural or man-made process [164]. Different hemophilic plant growth-promoting bacteria (PGPB) are prevalent in soil and aid in the alleviation of salt stress by encouraging vegetal nutrition and development via the methods depicted in Figure 4.

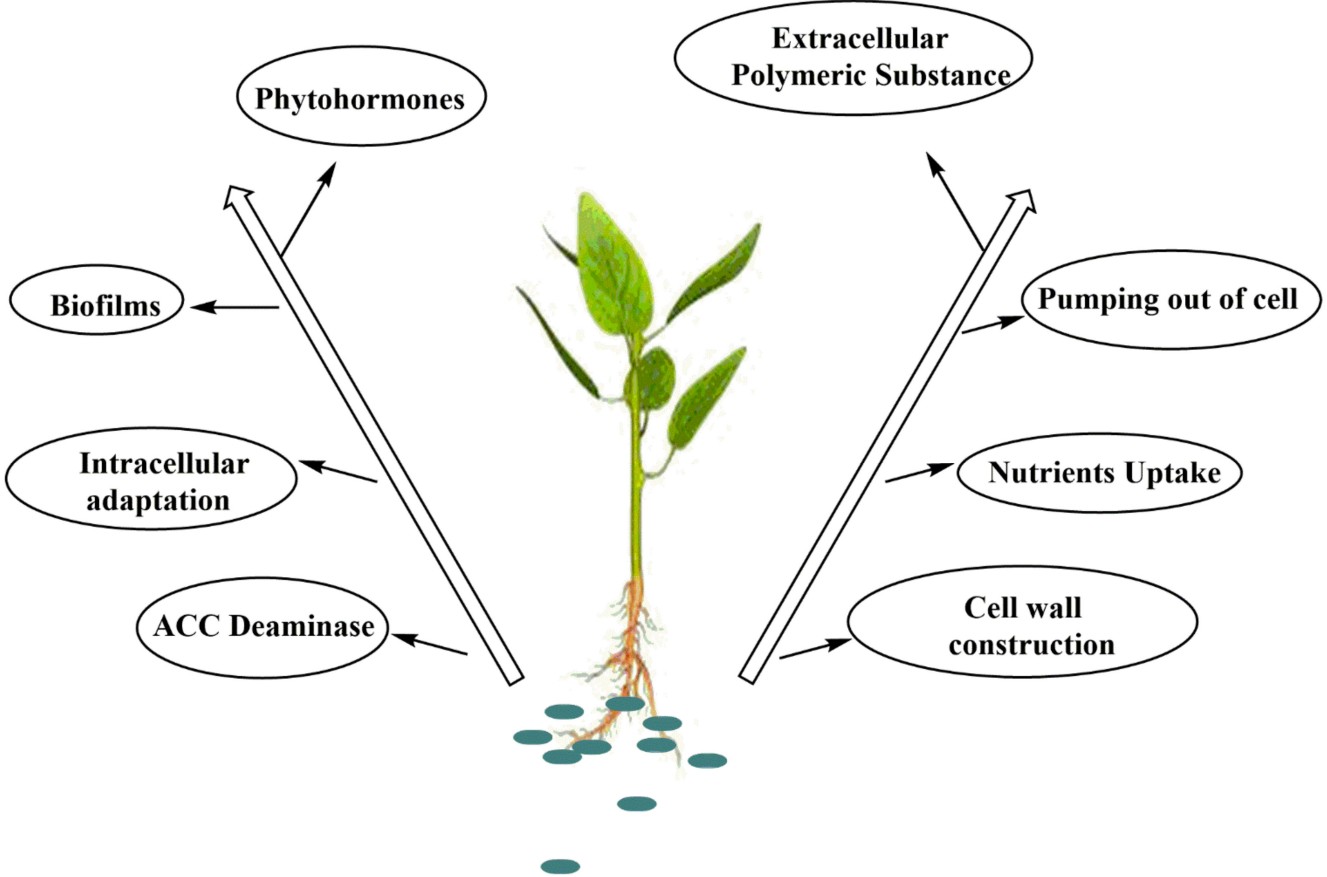

**(Plant Growth Promoting Bacteria) PGPB**

**Figure 4.** Different mechanisms of PGPR for plants developed under salt stress.

The PGPB suppress several plant pathogenic microorganisms and promote plant growth by many mechanisms, such as the production of various plant hormones, mobilization and the decay of organic material, along with an increase in the bioavailability of various soil mineral nutrients such as iron (Fe) and phosphorous (P) [165]. The PGPB produce plant hormones such as auxins and cytokinins that encourage the proliferation of shoots and modify the root system by the overproduction of roots and root hairs, which results in the improvement of water and nutrients uptake by the plants. Several bacterial species such as *Enterobacter* sp. encourage plant development via ACC deaminase activity, HCN production, siderophore production, IAA production and the solubilization of P [166]. Two bacterial species such as *Streptomyces rochei* and *Streptomyces sundarbansensis* produce IAA and encourage plant growth [167]. Soil is the host of a massive number of bacteria (usually between $10^8$ and $10^9$ cells per gram of soil); however, out of this, only 1% are culturable [168]. Bacterial genera such as *Streptomyces*, *Azospirillum*, *Clostridium*, *Alcaligenes*, *Bacillus*, *Rhizobium*, *Pseudomonas*, *Thiobacillus*, *Serratia* and *Klebsiella* are found to be effective as PGPB under salt-affected conditions [169]. The different field trials of

these species—inoculant or as a part of microbial consortia—have been found to be positive [170]. The PGPB have been found to be successful in maintaining osmotic potential, ion homeostasis and turgor potential, which helps in reducing the salt stress in plants [171]. The salt-tolerant microbial community helps in maintaining the health of salt-affected soil, sustains soil ecology and encourages the growth and development of plants [172]. Further research is required to determine the unknown mechanisms behind salt-tolerant microbial diversity [173].

### 8.1. Production of Phytohormone and ACC-Deaminase Activity

Plant growth-promoting bacteria are known to encourage the growth and development of plants by the synthesis of various plant hormones such as auxin, cytokinin, gibberellin and the minimization of ethylene by ACC deaminase. Ethylene is a well-known gaseous hormone that accumulates in plants under different abiotic stresses. The extent of ethylene buildup in plants varies by species and genus, as well as by organs and tissues. Ethylene is responsible for various development processes of plants such as the germination of seed, the development of root hairs, the ripening of fruits, the abscission of leaves and the senescence of plant parts by controlling various stress-related genes [174]. However, a higher accumulation of ethylene during stress conditions may become harmful for plant development [175]. The PGPB improve plant growth and help in salt tolerance by regulating the ethylene hormone level in plants through the ACC deaminase, which splits the ethylene precursor ACC to ammonia and $\alpha$-ketobutyrate, which results in the improvement of plant growth and fights salt stress [176]. The PGPB through ACC deaminase activity alter the surface area of the root and the number of root tips. Hence, the PGPB promote nutrient accumulation and the survival of plants under stress situations. It has been found that the synthesis of the enzyme ACC deaminase and the decrease in ethylene production are the main causes of PGPB-mediated plant growth promotion under salt stress [177]. Auxins are another class of plant hormones that can be regulated by PGPB. The auxins group includes I3B (indole-3-butyric acid) and IAA (indole-3-acetic acid); the bacteria producing the auxins group are *Actinobacteria*, *Nocardia*, *Frankia*, *Kitasatospora* and *Streptomyces*.

### 8.2. Production of Extracellular Polymeric Substance

Soil microbes synthesize various biopolymers such as polysaccharides, polyamides and polyesters under natural conditions. Along with this, wide spectra of polysaccharides are produced, such as structural, intracellular, extracellular or exo-polysaccharides [178]. Plant growth-promoting bacteria produce an extracellular polymeric substance that has an important role in mitigating salt stress [112]; this extracellular polymeric substance has the capacity to combine with cations such as sodium, resulting in a decrease in the bioavailability of these cations for plant uptake. The extracellular polymeric substance increases bacterial survival under salt stress conditions by improving the water-holding ability of soil and controlling the flow of soil organic carbon. The extracellular polymeric substance also aids in the formation of plant–microbe interactions [179] by giving a microenvironment where microorganisms can live under salt stress conditions. Root exudates support microbes in contacting the roots of plants and colonizing them. The amount and composition of extracellular polymeric substances change dramatically during salt and drought stress conditions. The extracellular polymeric substance is produced by microorganisms such as slime material which bind with soil particles by a Van der Waal attraction, hydrogen bond, cation linkage and anion adsorption phenomenon [176].

### 8.3. Production of Plant Osmolytes and Antioxidant Activity

Plant growth-promoting bacteria synthesize organic osmolytes such as sugars, glucosyl glycerol, alcohols, betaines, amino acids, tetra-hydropyrimidine, etc. [101]. Organic solutes found in the cytoplasm of bacteria can or cannot be synthesized by bacteria; sometimes the organic osmolytes are taken up from the outer environment [101]. The presence of this osmolyte helps to combat salt stress. The osmolytes produced in the cytoplasm help

in maintaining the osmotic balance of plant cells. Some common plant osmolytes include di- and oligosaccharides, betaine, proline, alcohols, glutamate and glycine [180]. Osmo protectants such as sugars and primary disaccharides such as sucrose and oligosaccharides such as raffinose and fructans are the basic drivers behind plant stress management. Sucrose production is connected with the survival of *Craterostigma plantagineum* during plant tissue dehydration [181]. During salinity stress, a higher fraction of the cellular energy deviated towards the formation of osmolytes is capable of defending the cells from osmotic fluctuations [182]. Osmolyte buildups preserve turgor pressure and balance the different macromolecular structures towards the physiological drought caused by salt stress [183].

### 8.4. Siderophore Production

The bacterial strains producing siderophores have a higher affinity for iron than phytosiderophores; therefore, they can remove Fe from the phytosiderophore complex. Researchers reported that the activities of microorganisms have a significant effect on the accumulation of iron in roots and its transport to other plant parts [184]. As reported by Rungin et al. (2012), the siderophore-producing endophytes increase plant root and shoot biomass because of the enhanced supply of iron. Siderophore-producing PGPB have been found to be successful in improving salt tolerance in the plant [185,186].

### 8.5. Induced Systemic Resistance

Induced systemic resistance (ISR) is the improved protection capability created by a plant against different types of plant pathogens succeeding in root colonization by microbes [187]. In addition to ethylene and jasmonate, other microbial substances such as pyoverdine, flagellar proteins, β-glucans, chitin, salicylic acid and cyclic lipopeptide surfactants have been found to operate as signals to stimulate systemic tolerance [188]. Plants create tolerance in response to pathogen and insect attacks and the colonization by microorganisms; however, this mediated condition is revealed by the stimulation of "dormant" immune responses reflected in the reaction to the external interactions of insects, pathogens and other invaders [189].

### 8.6. Essential Nutrient Uptake

Salt stress to plants reduces their nutrients uptake and accumulation capabilities such as N, P and K, along with their water uptake due to high osmotic potential and ion toxicity. Therefore, plants need more nutrients to survive in stress situations [190]. Crop yield is adversely affected in salt-affected soils because of the hindering nutrient uptake and translocation [191]. Plant-associated PGPB are well known for promoting water and nutrient absorption by plants [192]. The PGPB inoculation to plants increases nitrogen accumulation by a symbiotic and non-symbiotic relationship with the roots [193]. Phosphorus is found in organic and inorganic fixed forms in soils, and its major part is unavailable to plants. PGPB can convert these unavailable phosphorus forms into available forms by the mechanism of acidification and chelation processes [194]. Potassium is also an essential nutrient to plant growth; most of the K is found in fixed forms in the soil that are not available for plant uptake. Moreover, under salt stress conditions, the availability of potassium to plants decreases; under this situation, K-solubilizing bacteria (KSB) were found to be efficient in fulfilling the potash requirement of crops [195]. The K-solubilizing bacteria (KSB) group can convert mineral potash into available forms for plant uptake [196]. PGPB (Plant growth-promoting bacteria) enhance the availability of other essential elements such as copper, iron, manganese, zinc, etc. for the plants by the mechanism of chelation and acidification in soil [194]. Organic phosphate is resistant to mineralization. The microbial biomass is very important in the phosphorous cycle; microbes make it available for the plants [197].

## 9. Microbial Inoculation Influencing Soil Properties

Soil microbial diversity plays important role in improving soil health by controlling the supply of nutrients and the decomposition of OM, thus enhancing nutrient availability to plants. The production of different enzymes, hormones and macro-aggregates helps to sustain soil health. The salinity stress of soil drastically reduces the microbial diversity in the soil. Soil with good health conditions consist of around 600 million microorganisms in one gram of soil, with 15,000–20,000 distinct species, but the same amount is reduced to 1 million in salt-affected soil [198]. Salinity reduces microbial activity, microbial modification and OM degradation [199]. Furthermore, the different microbial groups in soils play a key role in the soil regulatory process for the nutrients cycling in salt-affected soil [200].

Fungal and bacterial abundance also play an important role in controlling soil respiration, which is a direct effect of microbial abundance on soil. However, changes in microbial abundance are also largely driven by soil properties [201]. The specialized soil functioning (e.g., denitrification) relies on specific groups of micro-organisms and is highly dependent on bacterial community composition [202]. The cementing properties of exopolysaccharides (EPS) strengthen the aggregate formation of the bacteria with the soil particles and bind Na ions, thereby reducing their toxicity in the soil. Therefore, a higher population of EPS-producing bacteria in the root zone will reduce the concentration of $Na^+$ available for uptake, thereby alleviating the salt stress effect on the plants [203].

## 10. Future Challenges for Salt Stress Mitigation through Halophilic Microbes

The identified halophilic plant growth-promoting microbes needs to be applied in agriculture to enhance crop yields under salt stress conditions. The development of biological products based on beneficial halophiles can extend the range of options for maintaining the healthy yield of crops in salt-affected soils. In recent years, a new approach has been developed to alleviate salt stress in plants by inoculating crop seeds and seedlings with salt-tolerant plant growth-promoting microbes. Thus, there is a great opportunity for halophilic PGPR's successful application in agriculture. The microbial formulation and application technology are crucial for the development of commercial salt-tolerant bio-formulation effective under salt stress conditions. Bio-formulations offer an environmentally sustainable approach to increasing crop production and health. Apart from microbial reclamation, improving the fertility of salt-stressed soils is another aim to be focused on. It has been observed that inoculation with mixed strains is more consistent than single-strain inoculations. Studies on the detailed mechanism of mycorrhizal fungi-associated plant growth under salt stress are lacking, and this needs to be explored. The promising approach toward tackling the problem of soil salinity by utilizing beneficial microorganisms including halophilic PGPR will make the greatest contribution to the agricultural economy, as they provide a cheap and eco-friendly approach to mitigate salt stress.

One of the recent focuses of research involves the application of PGPR to combat salt stress. The development of biological products based on beneficial microorganisms can extend the range of options for maintaining the healthy yield of crops in saline habitats. In recent years, a new approach has been developed to alleviate salt stress in plants by treating crop seeds and seedlings with PGPR. The great opportunity for salt tolerance research is its ability to be combined with halophilic PGPR.

The bottom line of every inoculation technology is its successful application under agricultural and industrial conditions. The microbial formulation and application technology are crucial for the development of commercial salt-tolerant bioformulation that is effective under salt-stress conditions. Bioformulations offer an environmentally sustainable approach to increase crop production and health, contributing substantially to making the twenty-first century the age of biotechnology. Apart from bioformulation, reclaiming and improving the fertility of stressed sites is another aim to be focused on. The promising approach toward tackling the problem of soil salinity by utilizing beneficial microorganisms including PGPR will make the greatest contribution to the agricultural economy—if inexpensive and easy-to-use stress tolerant strain formulations could be developed.

## 11. Conclusions

Despite the overall growth of salt-affected soils and the challenges associated with their reclamation and management, this review assessed current information on salt-affected soils and their bioremediation using various microbial techniques. The microbial consortium is playing an essential role in the long-term growth of agriculture. Similarly, halophilic bacteria can aid in the reduction of salt-stressed soil. So, plant growth-promoting rhizobacteria that thrive in salt-stressed soil must be employed in salt-affected soil for long-term growth and development. The development and application of halophilic plant growth-promoting microorganisms may aid in the long-term production of crops under salt-affected soil conditions. The creation of favorable halophile microbial formulations can broaden the number of choices for sustaining the crop output in salt-affected soils.

**Author Contributions:** Conceptualization the article structure and content: C.K., A.K., J.P. and A.P.; Defined the literature search criteria: A.P., C.K., J.P., P.D. and A.K.; Data handling, tables, and figures preparation: C.K., J.P., A.K., A.P., S.S.S. and P.D.; Writing–original draft: A.P., C.K., S.S.S. and P.D.; Writing–review and editing: A.P., C.K., A.K., J.P., S.S.S., P.D., G.K.Y., S.K.D., R.V. and G.L.K. All authors have read and agreed to the published version of the manuscript.

**Funding:** This research received no external funding.

**Institutional Review Board Statement:** Not applicable.

**Informed Consent Statement:** Not applicable.

**Data Availability Statement:** The data and materials will be made available on from the corresponding author(s) upon reasonable request.

**Conflicts of Interest:** The authors declare no conflict of interest.

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
