# Peer review of "Microbial Diversity and Adaptation under Salt-Affected Soils: A Review"

_sustainability, doi:10.3390/su14159280_

Round 1

Reviewer 1 Report

Microbial mediated remediation of Salt Affected Soils: a review

  1. Typos, spaces, and uppercase issues are found throughout the manuscript.
  2. In abstract, In India nearly 6.74 million ha-1 areas of what?
  3. Please rework and improve the graphical abstract
  4. In Table 1, 2021 is cited. Please cite recent 2021 and 2022 papers.
  5. Figure 1 can be improved. The fonts are not uniform in style.
  6. In subsection 5.4, please provide the genus names for the PGPR and provide a brief mechanism on how PGPR can enhance the expression of antioxidative enzyme activity.
  7. In 5.5, provide an example of a halophilic strain that produces siderophores.
  8. Italics scientific name in section 6.
  9. Improve table 2.
  10. On page 12, in the table, in AMF fungal species, there is nothing called Glomusm.
  11. In section 7, what will be the impact on the strain that produces cyanotoxin?

Reviewer 2 Report

The manuscript is a literature review which focused on, among others, the effect of soil salinity on microorganisms soil community and interaction of plants and microbes in salt-affected soils.

1. The article should be carefully checked for: the separation of connected words, spaces and punctuation marks. 
2. In my opinion the title does not reflect the content of the article. The title suggests that there will be a discussion about the remediation of various compounds in soils characterized by high salinity.
3. "The bacterial hormones which are the plant growth hormones produced in the vicinity of the plant roots can start a physiological response in the host plant"- please explain what bacterial hormones are and what functions they perform in their cells?
4. ha-1 - what unit is it ? (e.g. in sentence "..6.74 million ha-1 areas are occupied by salt-stressed soil.."
5. All the names of the bacteria should be written in italics.
6. "Nitrogen recycling by remineralization and immobilization by playing microbial response ..." unclear please explain; is it about nitrogen compounds? what type? 
7. Mechanisms of adaptation in saline conditions by halophilic bacteria shoul be more explain. 

Reviewer 3 Report

Review to

Title: Microbial mediated remediation of Salt Affected Soils: a review

Authors: Chiranjeev Kumawat, Ajay Kumar, Jagdish Parshad, Shyam Sunder Sharma, Abhik Patra, Prerna Dogra, Govind Kumar Yadav, Sunil Kumar Dadhich,Rajhans Verma, Girdhari Lal Kumawat.

Journal: Sustainability

Manuscript number: sustainability-1661291

General remarks: The manuscript by Kumawat is an interesting review focused on the effects of microbial communities able to counteract the effects of an excessive salinity in soil. In my opinion the manuscript should be restructured avoiding too scholastic information and repetitions. Authors should focus the contexts on the effective use of microbial remediation of salinity, which is the original purpose of the authors (as described in the abstract). The present version of the manuscript is too dispersive.

Another important point is that, the contents of the manuscript are very interesting and potentially useful but the manuscript is not appropriately written. The manuscript is full of typing and spelling errors. An extensive improvement of the English grammar and fluency is mandatory for the final acceptance. Based on these evaluation this manuscript will be suitable for publication on sustainability after substantial revision.

Major points

Please improve quality of graphical abstract

The introduction is too much generic in some parts, and should be better developed, e.g. detailing the effective influence of salt stress on crops and describing the main species affected by salinity. India would be an excellent study case.

It is not clear if arrows in Table 2 give qualitative or quantitative information. In the second case, authors should clarify this in captions. Further, the arrows should be similarly drawn (e.g. same dimensions, same distance from border etc).

Authors should improve quality of figures 2-3-4.

Minor points

Authors mentioned only 27 papers (out of a total of 194) from 2018-22. Authors should re-check bibliography.

Authors should pay attention to punctuations typing and spelling errors.

Reviewer 4 Report

See attached file

Round 2

Reviewer 1 Report

Thanks for improving the manuscript.

Reviewer 2 Report

The authors referred to the suggestions. Manuscript can be accepted in present form.

Reviewer 3 Report

The R1 version of the manuscript by Kumawat et al. is significantly improved. The requested revisions were provided by the authors. In my opinion is now suitable for publication on Sustainability.